# Impact of Sensory Afferences in Postural Control Quantified by Force Platform: A Protocol for Systematic Review

**DOI:** 10.3390/jpm12081319

**Published:** 2022-08-16

**Authors:** Joanna Aflalo, Flavien Quijoux, Charles Truong, François Bertin-Hugault, Damien Ricard

**Affiliations:** 1Université Paris-Saclay, CNRS, ENS Paris-Saclay, Centre Borelli, 91190 Gif-sur-Yvette, France; 2Université Paris-Cité, CNRS, Centre Borelli, 75006 Paris, France; 3ORPEA Group Research Department, ORPEA Group, 92800 Puteaux, France; 4Service de Neurologie de l’Hôpital d’Instruction des Armées de Percy, Service de Santé des Armées, 92140 Clamart, France; 5Ecole du Val-de-Grâce, Service de Santé des Armées, 75005 Paris, France

**Keywords:** posturography, gerontology, balance, sense organs, falls

## Abstract

Older adults’ postural balance is a critical domain of research as balance deficit is an important risk factor for falls that can lead to severe injuries and death. Considering the effects of ageing on sensory systems, we propose that posturographic evaluation with a force platform exploring the effect of sensory deprivation or perturbation on balance could help understand postural control alterations in the elderly. The aim of the future systematic review and meta-analysis described in this protocol is to explore the capacity of older adults to maintain their balance during sensory perturbations, and compare the effect of perturbation between the sensory channels contributing to balance. Seven databases will be searched for studies evaluating older adults’ balance under various sensory conditions. After evaluating the studies’ risk of bias, results from similar studies (i.e., similar experimental conditions and posturographic markers) will be aggregated. This protocol describes a future review that is expected to provide a better understanding of changes in sensory systems of balance due to ageing, and therefore perspectives on fall assessment, prevention, and rehabilitation.

## 1. Introduction

Older adults’ postural balance is a critical domain of research as balance deficit is an important risk factor for falls [1] that can lead to severe injuries and death. Although most falls are non-fatal, they are the second leading cause of unintentional injury deaths worldwide, and an estimated 37.3 million falls are severe enough to require medical attention each year, according to the World Health Organization [2]. Adults over 60 years of age are the most at-risk population [2]. Beyond the direct and traumatic consequences of a fall, these incidents of everyday life can lead to reductions in physical activity or even psychomotor disorders favoring the appearance of fragility or dependence, which can quickly become established and last. Preventing falls, and therefore ultimately slowing the emergence of dependency induced notably by a reduction in physical activity, requires high-performance screening tools, i.e., tools sufficiently sensitive to the detection of people requiring special care, but also sensitive to changes in sensorimotor capacities that can have a positive or negative impact on the risk of falling. Beyond the assessment of physical abilities, a screening tool should also be reliable. In the context of measuring balance abilities, this means that the tool must provide repeatable, reproducible, and sufficiently accurate measurements.

Age directly impacts the risk of falling and the correlation between the two has been extensively established [2]. The maintenance of balance involves many neurological systems (both in sensory integration, which involves sensory systems to record visual, somatosensory, and gravitational information via the vestibules, but also in the processes of decoding the information at the level of the spinal and supraspinal centers, which will give meaning to this information) [3] and muscular systems, mainly through the coordinated response to destabilization [4]. In summary, sensory integration at the level of the peripheral and central nervous systems induces a weighting of the information coming from the different channels, followed by contextualization. Finally, a motor response is produced to induce a postural adjustment as well as a focal movement [5] itself compared to the desired movement according to feedback loops that are particularly effective in keeping one’s balance in spite of a biomechanical structure with many degrees of freedom and a significant redundancy in the muscular actions that can be induced [6]. This optimization of sensorimotor processes, of which a significant part is acquired from the experience of individuals [7], will then be impacted by degenerations linked to age and inactivity.

Static steady-state balance refers to ability to maintain a steady position while sitting or standing [8]. Thus, an adequate postural control relies on the integrity of afferent and efferent pathways, as well as sensorimotor reflex capacities. Afferent systems contributing to postural control include the visual, somatosensory (i.e., tactile and proprioceptive) and vestibular systems, while efferences refer to muscular strength and resistance, and joint mobility. Ageing can affect all of these systems, as well as sensorimotor reflexes, in terms of precision or reaction time, resulting in postural control deficits [9].

In order to detect these deficits, clinicians can test the efficiency of each system involved. The integrity of sensory receptors and pathways contributing to postural control can be assessed by specific tools: discrimination tests for visual and somatosensory inputs, position and movement discrimination and reproduction for proprioception [10], and vestibular test battery for gravitational sense [11].

Another approach is testing the function with balance tests. Static steady-state balance is usually a subtest of scales also evaluating dynamic balance. Among the most commonly used tools, we can cite the Berg balance scale, the Short Physical Performance Battery and the Tinetti Performance-Oriented Mobility Assessment [12]. However, due to redundancy in sensory information and potential compensatory mechanisms, such tests are not able to detect deficits that could lead to an increased risk of falling in specific situations. In order to tweak balance function evaluation, further exploration of neurological and sensorimotor systems can be performed by modifying the sensory conditions during the evaluation. For example, the inability for a subject to maintain steady balance after closing their eyes can reflect a dysfunction of proprioceptive receptors or pathways [13]. In the same way, subjects with visual or vestibular deficits, who are dependent on somatosensory input, will exhibit an excessive loss of balance when standing on a foam or unstable surface [14,15]. Such evaluation protocols are cheap, easy, and quick to administer, but suffer from various limitations: ceiling effect, inter-rater reliability, or lack of sensitivity to change [16].

However, the development of technology and computerized tools, such as force platforms, capable of recording proxies of the neurophysiological mechanisms that are involved in the maintenance of balance has benefited to clinical assessment of postural control and fall risk [17]. Without directly recording the action of the neurological and muscular systems in maintaining balance, the recording of the displacement of the center of pressure (CoP—the point at the surface of the force platform where the resultant of the ground reaction forces applies), through the quantification of ground reaction forces during bipodal stance, provides indications of the stable or unstable state of the participant. Indeed, the complexity of postural control, even for a relatively simple task such as bipodal standing, makes it necessary to use interpretable indicators, which are partially representative of balancing abilities, but can be used to longitudinally follow patients regarding their risk of falling.

In this respect, computerized posturography was first used in the 1970s as a new way to quantify balance using a force platform [18]. The CoP and its displacements during a steady-state task are used to represent a subject’s postural sway. The analysis of a subject’s postural control based on a posturography relies on features (i.e., global variables) extracted from the COP trajectory during recording, such as the mean or maximal amplitude or velocity, the mean frequency, or the ellipse area covering the signal [19]. A 2-dimensional projection of the CoP trajectory (i.e., statokinesigram, see Figure 1) can also be generated from the recording for visual analysis, which can be useful for clinical applications as it is a faster than extracting and analyzing CoP features.

Compared with clinical tests, computerized posturography seems to be more objective and reliable since it relies on an automated extraction of features, rather than a scoring system [20]. This tool proved to be useful to discriminate groups of patients with balance disorders. For example, features such as COP path length, COP velocity, and sway area (95% confidence ellipse area) during quiet stance are significantly greater in people with multiple sclerosis [21] or diabetic neuropathy [22] compared with healthy subjects.

The relevance of this tool for assessing fall risk in healthy older adults is a topic for scientific research, and the current evidence has been reviewed in several publications. In a systematic review focusing on follow-up studies using posturographic assessment to predict falls, Piirtola and Era [23] found several CoP features derived from static balance associated with future falls across the studies analyzed. Only one CoP feature was significant in both eyes open and closed conditions: the mean sway amplitude in the mediolateral (ML) direction. The mean velocity of CoP in the anteroposterior (AP) direction was significant only with eyes open, while the CoP velocity and the root-mean-square values for CoP displacement in the ML direction were significant only with eyes closed.

In a narrative review of the literature, Pizzigalli et al. [24] cited similar features related to ML displacement that were significantly different between fallers and non-fallers in all conditions. Again, the significance of some other features varied with sensory conditions, for example, in Melzer et al. [25], a study included in this review, the sway area was a discriminating feature only in the eyes closed and foam conditions. Regarding the methodology of the studies, the authors pointed out the variety of protocols and measurement methods, advocating for harmonization. These conclusions were confirmed in our previous literature review and meta-analysis [26], highlighting the relevance of static posturography in identifying elderly people at high risk of falling; however, there is a need for different protocol and analysis methods to improve its predictive value. Indeed, further exploration is needed to identify the most relevant recording protocols and CoP features, as well as the neuro-biomechanical processes underlying the relationships between postural control and falls. However, one interesting finding across the reviews cited above is the impact of sensory conditions on the significance of CoP features. However, this finding and its potential to improve the predictive value of posturography through protocols exploring such sensory conditions in detail are rarely discussed. 

Computerized posturography allows for similar protocols as clinical tests, including complex manipulations of sensory conditions to explore a subject’s ability to adapt to sensory perturbation or deprivation. This type of protocol was found to be valid and reliable to detect postural control changes in people with multiple sclerosis [27] or able to differentiate people with progressive supranuclear palsy, Parkinson’s disease, and healthy subjects [28] based on balance index for each sensory channel tested. However, this technology also has limitations: it requires more expensive materials than clinical balance scales, as well as experience to infer clinical information and implications from posturographic features or visual analysis.

In this regard, and considering the effects of ageing on sensory systems involved in maintaining balance, we propose that posturographic protocols exploring older adults’ sensory organization and adaptation in various conditions could help to specify postural control alterations, and therefore fall risk assessment. To our knowledge, there is currently no published work reviewing the literature on this topic. For this reason, the systematic review described in this protocol will focus on studies exploring the ability of the elderly to adapt to different sensory conditions during quiet stance, through computerized posturography.

## 2. Objective

The objective of the review is to collect and analyze publications comparing the impact of sensory perturbations on older adults’ static balance, with respect to younger subjects, using a force platform.

## 3. Research Questions and Hypotheses

The primary question this systematic review and meta-analysis protocol was designed to address is:When exposed to sensory deprivation or perturbation during quiet stance, are older adults able to maintain their balance or do they exhibit an increased instability compared with younger subjects? The primary criteria of this review will therefore be the variability of posturographic markers (i.e., CoP features), and our hypothesis is that older adults exhibit an increased instability, with respect to younger subjects, when exposed to sensory perturbations.

In order to better understand the relationships between sensory systems and postural control, we will set two secondary research questions: What is the impact of experimental sensory conditions on the balance of elderly subjects? For this question, we will compare the impact of the perturbation or deprivation of each sensory channel on CoP features. Based on the literature cited above [24], we expect that visual perturbations will have a greater impact on older adults’ balance, compared to proprioceptive and vestibular perturbations.Which features of CoP displacements are used to assess the sensory organization of postural control during quiet stance in the elderly (≥60 years)? To address this question, we will extract the list of the CoP features assessed in each study included.

## 4. Methods

### 4.1. Research Protocol

This literature search and analysis was designed according to the PRISMA (Preferred Reporting Items for Systematic Reviews and Meta-Analyses) updated guidelines [29]. This protocol was registered in the PROSPERO database under the ID CRD42022309566.

### 4.2. Search Strategy

An electronic database search of titles and abstracts published will be performed between February and July 2022 to identify all articles published that include posturography during quiet stance under various sensory conditions in older adults.

Six databases will be used as sources for published articles: Medline (PubMed), Cochrane CENTRAL, ScienceDirect, Web of Science, Scopus, and BDSP. The search will be performed for articles published before February 2022, using associations of relevant keywords, following the PICO methodology (see Table 1). The keywords «Vision, Ocular», «proprioception», «touch», and «postural balance» will be used as MeSH terms when possible. The main database search will be supplemented by a review of grey literature, which will be conducted through web searches on Google Scholar and Biosis. Two clinical trials registry platforms will also be searched: ICTRP and ClinicalTrials.gov. In addition, all reference lists and bibliographies of included studies will be reviewed for relevant studies that were not picked up through the electronic search.

### 4.3. Inclusion and Exclusion Criteria

Randomized control trials (RCTs), non-randomized control trials, and observational studies will all be eligible for inclusion. Due to the risk of bias arising from only including data from published RCTs [30], data from grey literature will also be included provided that they meet the inclusion criteria (Table 2). Exclusion criteria will also be set regarding the type, publication date, and the language of the article, as well as the age of the population studied (see Table 3).

In terms of the research area of the studies we wish to select, we will focus on the exploration of elderly people’s postural control with and without sensory deprivation or disturbance, but that seek a comparison between a control situation and a situation with altered sensory afferences. This therefore includes studies that explore balance through a force platform with the addition of a sensory altering tools, which includes, but is not limited to, eye closure, gaze stabilization devices that limit identification of the visual vertical, blindfold tools and masks, screens to provide a visual environment or create a sensation of vection (i.e., an illusion of movement) as can be performed with a large screen or virtual reality, etc. This list of examples presents only some of the possibilities of altering visual afferences during a balance test, and does not reflect the completeness of the methods of altering sensory inputs, for the purpose of illustrating the methodologies being sought. To this we can add other devices and techniques for other sensory systems: vestibular (i.e., galvanic), proprioceptive (i.e., tendon vibration), tactile (i.e., foam pad) and even auditory if this appears in our bibliographic research. However, this research does not include protocols that would involve a dynamic approach to balance measurement (e.g., assessment of anticipatory muscle activities, walking, target search with biofeedback, unipodal tests, as well as external or examiner-generated destabilization…). Regarding this last point, mobile force platform tools and technologies that serve to reduce the perception of movement through servomotors controlled by body oscillations will be included.

### 4.4. Paper Review Process

Potentially eligible studies will be screened for inclusion eligibility independently by two review authors (FQ and JA) based on their title, abstract, and full text. Potential disagreements on inclusion eligibility will be resolved by consulting a third reviewer (DR).

Articles will first be imported into the Zotero^®^ bibliographic database (Corporation for Digital Scholarship and the Roy Rosenzweig Center for History and New Media, USA) before screening so that all articles can be reviewed from the same source in order to select those that meet the criteria. If there is disagreement between the reviewers, the study will be discussed until a consensus is reached. Papers that are eligible will then be subjected to data extraction and a “risk of bias” evaluation, as described below.

### 4.5. Risk of Bias Evaluation

Following Cochrane’s handbook guidelines [29], an individual quality/risk of bias assessment will be performed by using a 27-item checklist (see Appendix A, Table A1) based on the Single-Case Reporting Guideline In BEhavioural Interventions [31]. The descriptions have been modified to be more specific to interventions using measurement instrument, following recommendations from the COSMIN risk of bias tool [32]. One item (9—blinding) from the original guidelines was deleted since it is not relevant to posturographic evaluations. Two original items about procedural fidelity (6 and 17) were merged into one. For item 18 (analyses), considering the topic of this review, involving computerized tools and quantitative analyses, we decided that it was important to assess all stages of analyses for a more precise evaluation. Therefore, it was divided into four items: data pre-processing, analyses and reporting, as well as potential data dredging. Finally, we set a scoring system for each item, to allow a maximum score of 27 for a study meeting all the criteria. Quality assessment for each article will be performed by two assessors (FQ and JA), who will be blind to the score given by the other assessor until both have completed the evaluation. Any disagreement over the final score for each article will be discussed; if no agreement can be reached, the rounded mean of both scores will be used.

### 4.6. Data Extraction and Analysis

Following inclusion of the articles for analysis, the text from each reference will be imported into Microsoft Excel (version 2016, Microsoft Corp., Redmond, WA, USA) for data extraction. One assessor (JA) will extract and collate information following Taylor et al.’s guidelines [33]. Another assessor (FQ) will verify the extracted data from the included articles in order to confirm coherence of the data. Key characteristics to be extracted will include information about the study itself such as author(s), title, year of publication, inclusion and exclusion criteria, sample size, study methodology. Population characteristics will also be extracted, including demographic and biometric data such as participants’ gender, age, weight, height, BMI, and cognitive capacities (e.g., following a Mini Mental State Examination (MMSE)). Quiet standing test parameters will include conditions of the tests such as the foot and body position (comfortable or standardized), the number and duration of trials, posturographic materials and settings. For the test itself, data will be collected on every feature for each sensory condition tested (e.g., varying visual surrounding, type of standing surface, with or without tactile or vestibular stimulation).

When these data are unavailable from the main text, additional files will be examined for more information. When data on the force platforms or other kind of equipment (such as the materials used to change sensory conditions) remain unspecified, information will be sought from other articles by the same author(s).

Regarding the results, for experimental studies, the available posturographic data will be extracted from the baseline measurements as long as they report sensory conditions during recording. If the baseline data are not included, the article will not be analyzed. When comparing different sensory conditions, if no information is provided regarding a sensory channel, we will consider it was tested with standard/baseline settings. For example, if perturbed somatosensation is tested and referred to as “foam condition” with no details regarding the visual afference, we will consider it as “foam and eyes open”. Data from control younger subjects will also be extracted as a control group when available. The data and experimental conditions will be extracted in a similar way for both groups, the elderly group and young, healthy, control participants.

Finally, authors will be contacted via email to request missing data when they are not available in the main text, additional files, or from other sources as described above. Extracting data based on figures, manually or with software, shows poor inter-rater reliability [34]. In order to avoid introducing bias, it will not be used to obtain missing data.

### 4.7. Strategy for Data Synthesis

Extracted data from included articles will be presented descriptively and will include study characteristics, experimental conditions, and posturographic features. Individual risk of bias will be assessed using the value of the percentage scores from the 27-item checklist. Score distribution will also be studied to look for a Gaussian distribution or, alternatively, a trend in favor of the studies included in the meta-analysis.

In order to pool results, at least two studies must have used the same posturography feature in similar experimental conditions. If the included studies show consistency between their protocols, particularly with regard to the homogeneity of patient populations and the experimental conditions, a meta-analysis of the aggregated data will be considered, following the Cochrane Collaboration handbook recommendations [35]. Means and standard deviations (SD) of measures, as well as the number of participants per group, will be used to compare the effect size of each condition on the postural stability and to allow the creation of forest plots. If SD data remain unavailable, even after contacting the authors, but standard errors or confidence intervals are available, we will calculate standard deviation values.

For features that cannot be aggregated into a meta-analysis, a “best evidence synthesis” method will be preferred, evaluating the strength of the studies’ evidence in regard to their score in the risk of bias assessment, with particular attention on the methodological quality of the studies.

### 4.8. Confidence in Cumulative Evidence

Sensitivity analyses will explore the impact of studies’ characteristics and recording settings on the features results during the quiet standing measurement. These settings include the body position, foot position, and recording duration, as well as population subgroups (e.g., frail, pre-frail, or healthy). The outcome variability due to the range and mean age of the group tested will also be discussed. If the heterogeneity for a given parameter within the meta-analysis is too high (as measured by *I*^²^ > 50%), we will explore the impact of deleting studies with particular settings (different materials, population, or body position, for example) in order to decrease the heterogeneity. We will then discuss the changes in heterogeneity in relation to the study/ies deleted and their settings.

If enough RCTs and interventional studies can be included, the overall quality of the evidence for each outcome will be presented using the GRADE (Grading of Recommendations, Assessment, Development, and Evaluation) criteria as per the Cochrane Collaboration [36]. GRADE’s approach to assess quality of evidence is based on eight criteria: study design, consistency of results, directness of evidence, precision-based on the optimal information size (OIS), magnitude of effect, effect of plausible residual confounding, dose response, and publication bias. The “dose response” criteria, irrelevant in this context, will be omitted. Otherwise, the cumulative evidence will be assessed using a scale based on our previous work [37], classifying the cumulative evidence as “high”, “moderate”, or “low” based on the risk of bias checklist mean score, number of studies, heterogeneity and cumulative sample size (see Table 4). Risk of bias mean scores were adjusted to fit the 27-item checklist, as well as the number of studies and cumulative sample size in order to be more specific to the exploratory design that we expect to encounter in posturography. We expect to have a smaller number of studies with similar experimental conditions and CoP features, while the sample size could be lowered due to the matched sample comparison rather than groups comparisons in the original publication of the scale [37].

To visualize possible publication bias, funnel plots will be used to represent the estimated effect size of each article against the standard error of the mean plotted on the vertical axis. A symmetric inverted funnel shape suggests no publication bias. A funnel plot will be drawn for each feature with respect to the sensory conditions explored. Studies using specific recording settings will not be included in the funnel plot.

## 5. Discussion

This protocol of a systematic review, collecting and analyzing previous results with evidence-based guidelines, aims to explore older adults’ sensory organization for postural control with a broad scope of experimental conditions. We choose to collect studies using posturography to evaluate static balance in the standing position because it is a widespread method, adapted to frail patients, and now affordable using cheaper force platforms. Our hypothesis is that posturography under different sensory conditions can highlight different strategies for postural control, and detect inabilities to adapt causing falls in elderly people with or without pathologies. This hypothesis is supported by previous work from Peterka and Loughlin [38] exploring the impact of sensory perturbations on the time–frequency distribution of body-sway velocity, suggesting the possibility that posturographic analysis could reflect differences in sensory reweighting strategies. Cohen et al. [39] reported another possibility for posturography: exploring the motor control strategy (i.e., hip or ankle-based strategy) based on the amplitude of shear forces generated by a subject, compared with normalized theoretical values. When swaying forward or backward, a subject can move back to a centered position with restoring forces derived either from torque at the ankle (ankle strategy) or from torque at the hip (hip strategy). These two strategies are characterized by specific sway trajectories [40], that could be identified with posturographic features in order to assess whether a subject can adapt their motor strategy depending on constraints. As motor response is dependent on sensory environment [41] and older adults exhibit specific muscle sequences [42], exploring those adaptive capacities for motor strategies in regards to sensory conditions could help specify the balance deficits due to ageing.

The different experimental conditions we can find in the literature are usually designed to test a particular sensory channel and therefore explore its contribution to maintaining balance. As a result, many conditions can appear to attest the impact of the modification or reweighting of a particular sensory input. Clinical applications include the sensory organization test (SOT), which tests visual, proprioceptive, and vestibular inputs under six different conditions by shunting the other sensory channels one by one. The SOT is an equipment-intensive test and uses well-defined conditions, especially to ensure reproducibility in the automatically calculated results from the center of the pressure trajectory. The “subtests” are therefore administered one by one under the prism of a static balance analysis. What we mean by this is the absence of dynamic exploration of sensory reweighting when subtracting or adding a sensory input. However, we found articles presenting more dynamic explorations, i.e., during the recording, of the modification of the posture following a modification of the information, as was the case for Eikema et al. [43], who recorded subjects’ CoP displacement pre-, during, and post-vibratory stimulation to the feet to disrupt proprioceptive input.

Our research will integrate these approaches, which are by nature more exploratory than validated clinical tests on a force platform, and we will aim to aggregate data from the literature if the experiments share sufficient similarities, both in the generation of alterations in sensory systems and in the analysis of data from the COP trajectory. This desire to aggregate data is particularly challenging in view of the varied possibilities of altering afferences on the one hand, but also in view of the use of new haptic or virtual technologies to alter senses, which do not necessarily allow sufficient hindsight on their innovative and therefore emerging uses [44,45,46].

Although we expect some protocols (such as eyes-open and eyes-closed) to be common, allowing us to aggregate the results, we expect a large diversity in other sensory conditions, materials, parameters or data pre-processing and analyses, as well as missing information. The methods described in this protocol, such as the thorough search for missing data, the extraction of several parameters regarding the conditions of recording, and the sensibility analysis, are meant to minimize the heterogeneity while avoiding irrelevant comparisons. The parameters such as the body or feet position and recording duration, that we decided to take into account, are based on previous publications exploring the influence of recording and anthropometric parameters on CoP features [47,48].

Limits for this systematic review include bias from the studies that will be included. Some of these biases were already identified in previous reviews [26], and the methods described in this protocol, such as the extended extraction of data on experimental settings described above, are intended to minimize them. The meta-analysis is meant to address inherent limitations of individual studies, such as small sample size and statistical power, while monitoring publication bias with funnel plots.

We decided to include studies with explicit posturographic features, as opposed to balance index with no information about the CoP features used to calculate it; this may resulting in potentially interesting studies getting excluded. However, the validity and reliability of a feature needs to be assessed, and the decision to keep explicit features will allow us to discuss their relevance to reflect postural control with regard to the current evidence.

Regarding the population, a potential limitation for the generalization of our future results sets in inclusion and exclusion criteria. When aggregating results, we will take into account the groups’ mean age and carry a sub-analysis stratified by subgroups (for example frail or healthy), when this information is available. However, our review will include studies about older adults with no major medical condition or medication, in order to explore the effect of age while avoiding confounding factors. While most studies set the same criteria, the actual population of older adults usually has one or several medical conditions, and medications, which were found to have an impact on balance and fall risk [49,50,51] for medications; [52] for medical conditions. This specific subject will therefore be discussed in the future review, since we believe it is an important factor to take into account when using results from research for clinical use.

Previous studies summarized the relevance of posturography and protocols involving varying conditions in different disorders affecting balance. A posturographic protocol with eyes open and closed during quiet stance was found more sensitive and accurate than a clinical test to predict falls in patients with multiple sclerosis [53]. In people with diabetes, the greater instability of those with sensory neuropathy at baseline was exacerbated by the association of visual deprivation (eyes closed) and vestibular perturbation (head back) [54]. A similar effect was found in people with vestibulopathy compared with healthy subjects when adding a foam pad to interfere with somatosensation [55]. For healthy older adults, the future review is expected to give the first insight of summed up evidence regarding the organization of sensory afferences for postural control.

Therefore, in order to define guidelines in the use of posturography in clinical evaluation, the future review will evaluate which protocols and parameters are useful to detect differences between sensory conditions. Furthermore, the data synthesis of relevant features is expected to improve our comprehension of changes in postural control due to ageing, such as sensory afference reweighting. Indeed, a potential deficit in sensory reweighting adaptation in older adults is still controversial. Several authors found older adults to have a slower and/or deficient sensory reweighting in response to proprioceptive [43,56] or visual [57] alterations, when compared to younger subjects. Another study [58] explored visual reweighting in younger subjects, healthy older adults and fall-prone older adults. While all older adults exhibited slower and longer sensory reweighting than young subjects, fall-prone subjects also demonstrated poorer capacities than their healthy peers in some conditions, depending on the stimulus amplitude. Conversely, Allison et al. [59] found no evidence of differences between younger and older subjects, either healthy or fall-prone, exposed to medio-lateral oscillatory visual inputs and fingertip touch. These contradictory results could be explained by the diversity of protocols, regarding the settings, the sensory channel/s disturbed, and the type of stimuli. Considering that many systems and functions are involved in postural control, further research is needed to better understand the conditions and modalities of sensory reweighting dysfunction due to ageing, and its potential impact on fall risk.

As fall is a multidimensional problem, combining intrinsic (i.e., biomechanical systems, sensorimotor integration…) and extrinsic factors (environmental characteristics) [60], the future review may not identify a single unique measurement method to discriminate the negative impact of ageing on postural control, leading to an increased risk of falling, but could provide insights into predictive markers related to an individual’s equilibrium capacities.

Such evaluation protocols, more detailed about the individual sensory organization, could also help with setting future orientations for patients’ physical activity or rehabilitation programs. Providing insights into an individual’s specific organization, for example dependence to a single sensory system, a posturographic evaluation of standing balance could alert the clinician on specific needs. Unequal reweighting of sensory afferences may suggest the need for rehabilitation or reinforcement regarding the underused sensory channels, or adjustments of the surroundings and aids to avoid falls. During rehabilitation, it could help setting balance programs aimed at improving sensory integration by manipulating environmental constraints [61]. Finally, posturography could be used to assess a program’s effectiveness based on the between-sessions comparison of the patient’s posturographic outputs, reflecting a patient’s improvement with greater sensitivity than clinical tests [16].

## Figures and Tables

**Figure 1 jpm-12-01319-f001:**
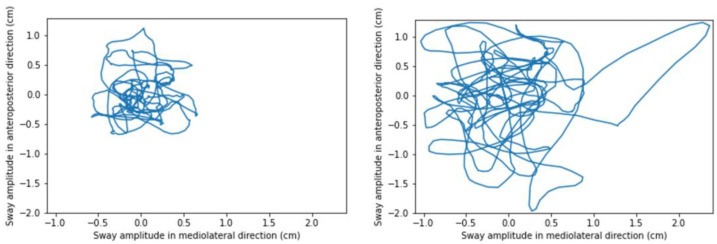
Example of a statokinesigram, representing the CoP trajectory of a subject with proprioceptive deficit, that we recorded standing with eyes open (on the **left**) and eyes closed (on the **right**) on a force platform.

**Table 1 jpm-12-01319-t001:** Keywords that will be used for each of the PICO domains. An asterisk (*) represents any group of characters and will be used to search for multiple variants of a word.

Population	Intervention	Comparison	Outcome
Older Elderly Community dwelling Nursing home Institutional care Assisted living facility Frailty	Quiet stan * Standing position Postural stability Posturography Force platform Statokinesigram Stabilogram Center of Pressure	Visual Vision Propriocep * Somatosens * Vestibular Tactile Touch Audit * Postur * sense Vection Motion illusion	Sensory organization Sensory preference * Sensory integration Postural sway Postural control Postural balance

**Table 2 jpm-12-01319-t002:** Inclusion criteria.

General criteria	Related to the main topic: “sensory organization during quiet standing in older people.” Articles not related to this topic will not be included based on the two-reviewer evaluation system.
Language	Articles written or translated in English or French.
Type of study	Clinical trials, randomized, or not. Observational, time series, and cross-sectional studies.
Participants	Older adults (aged ≥60 years) without a medical condition that could impact their posture.
Intervention	Articles analyzing balance during quiet standing under different sensory conditions, with a force platform. Articles analyzing static balance in any position other than standing, or analyzing dynamic balance, if they report static balance in standing position as a baseline measure.
Comparison	Articles will be included if they compare static balance under different sensory conditions such as: eyes open/eyes closed/perturbed vision; static visual surround/sway-referenced visual surround; static support surface/sway-referenced support surface.
Outcomes	Primary outcomes will be the features in the COP analysis used to compare postural control in the different sensory conditions.

**Table 3 jpm-12-01319-t003:** Exclusion criteria.

General criteria	Published after February 2022.
Type of article	Secondary sources such as literature reviews and meta-analyses.
Participants	Subjects with a medical condition that could impact their posture, including (but not limited to) Parkinson’s disease (PD), multiple sclerosis (MS), hemiplegia, paraplegic, stroke, or brain trauma. Orthopedic disorders affecting balance, such as recent arthroplasty or amputation, will also not be included in the review.
Intervention	Articles analyzing static balance in any other position than standing, or analyzing dynamic balance, without baseline measures in standing position. Studies analyzing balance with any other device than a force platform.
Outcomes	Studies with imprecise outcomes such as balance index with no information about how they are calculated or which CoP feature they’re based on will be discarded.

**Table 4 jpm-12-01319-t004:** Confidence in cumulative evidence scale that will be used when the GRADE is not appropriate.

Quality	Risk of Bias Mean Score on the 27-Item Checklist	Number of Studies (*n*)	Heterogeneity (*I*^²^)	Cumulative Sample Size
High	≥22	≥8	<30% (low heterogeneity)	≥200
Moderate	16–21	3–7	30–75% (moderate)	100–199
Low	≤15	0–2	>75% (high heterogeneity)	≤99
Score				

## Data Availability

Not applicable.

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
