# Peer review of "Impact of Sensory Afferences in Postural Control Quantified by Force Platform: A Protocol for Systematic Review"

_jpm, 2022, doi:10.3390/jpm12081319_

Round 1

Reviewer 1 Report

The protocol for systematic review written by Joanna Aflalo et al. aims to review the literature on the topic: Sensory organization during quiet stance in healthy older adults assessed by force platform. As the ability of elderly people to adapt to different sensory conditions and a constantly changing environment is crucial for preventing falls, I consider the topic of the future review as highly relevant and important for possible use in clinical practice, rehabilitation as well as research. I believe that this protocol could be also helpful for other authors who plan to prepare a review focused on a similar topic in the field of human balance control. I would like to encourage the authors to think about the next review exploring the assessment of postural balance and sensory organization using inertial sensors (i.e., accelerometers and gyroscopes).

I appreciate the nicely and consistently written Introduction providing adequate background of the future review´s topic. This section is easily readable, well-suited and sufficiently establishes the rationale for the future review. However, unification and correction of some of the basic terms are needed. Objectives are clearly formulated and defined. Dividing the section Methods into several sub-chapters with detailed content in each sub-chapter indicates that authors have very well thought out all the requirements that the future review must fulfill. I especially appreciate the summary of all keywords and criteria in the tables. Discussion is rich but still easy to follow including relevant works regarding the sensory organization in older adults.

As I mentioned above, the formal writing of the manuscript needs to be improved. Specifically, the authors should be more consistent in writing of specific terms and using it uniformly throughout the manuscript as well as be more specific in their statements. I will address all of my suggestions for each part of the manuscript in the specific comments in separately attached document.

Author Response

Thank you for your very detailed suggestions to improve this manuscript. Please see the attachment for our reply to your report. 

Reviewer 2 Report

Dear Authors,

I appreciate the author's efforts for witing this study. I recommend minor revision and have some comment as folllows.

1. Please indicate your hypotheses in the introduction.

2. Please add more details about GRADE criteria and its grading in the 'Confidence in cumulative evidence' section.

Author Response

Thank you for your report and suggestions that we followed. Please see the attachment for our reply.
